# Offline Policy Evaluation for Reinforcement Learning with Adaptively Collected Data

**Sunil Madhow**    **Dan Qiao**    **Ming Yin**    **Yu-Xiang Wang**
UC Santa Barbara
{sunilmadhow,danqiao,ming_yin}@ucsb.edu
yuxiangw@cs.ucsb.edu

## Abstract

Offline RL is an important step towards making data-hungry RL algorithms more widely usable in the real world, but conventional assumptions on the distribution of logging data do not apply in some key real-world scenarios. In particular, it is unrealistic to assume that RL practitioners will have access to sets of trajectories that simultaneously are mutually independent and explore well. We propose two natural ways to relax these assumptions: by allowing the data to be distributed according to different logging policies independently, and by allowing logging policies to depend on past trajectories. We discuss Offline Policy Evaluation (OPE) in these settings, analyzing the performance of a model-based OPE estimator when the MDP is tabular.

## 1   Introduction

Reinforcment Learning (RL) studies algorithms that learn to optimize a reward criterion by dynamically interacting with an environment. By balancing exploration with exploitation, *online* RL algorithms can very effectively learn to behave optimally. However, this mode of randomized exploration precludes some crucial use-cases for RL, such as for running medical trials, or training self-driving cars. Even in (comparatively) low-stakes applications like advertising, adopting elementary RL algorithms would mean throwing away vast reserves of data gathered from previous strategies. Both of these problems would be partially addressed by practical methods for *offline* learning, which seeks to solve the control problem using existing data.

In the offline RL setting, the goal is to perform RL tasks using existing data, $\mathcal{D}$, generated by some logging policy, $\mu$, and MDP $\mathcal{M}$. In Offline Policy Evaluation (OPE), we seek to estimate the value of a target policy $\pi$ under $\mathcal{M}$. In Offline Learning (OL), the goal is to use $\mathcal{D}$ to find a good policy $\pi \in \Pi$ where $\Pi$ is some policy class.

Practically, OL would allow RL practitioners to exercise more control over the gathering of training data. However, in order for $\mathcal{D}$ to be a rich enough dataset to learn from, strong assumptions need to be made about the exploratory properties of $\mu$. This is in addition to the already strong assumption that the set of trajectories is i.i.d. In Section 3, we discuss the problems that such assumptions pose and propose ways of weakening them. We believe the most fruitful of these to be Adaptive OPE (AOPE), where we allow each trajectory to be distributed according to a different logging policy, which may depend on previous data.

Some scenarios that AOPE covers and OPE does not are presented below.

1. The dataset $\mathcal{D}$ has been collected over a long period of time, during which unrecorded changes have been made to the policy. An example of this would be the portfolio of a longterm client at an investment company, which will certainly change over time.

Offline Reinforcement Learning Workshop at Neural Information Processing Systems, 2022

2. The dataset $\mathcal{D}$ was gathered by humans, and therefore influenced by a number of unobserved factors. For example, a doctor prescribing medicine may make a determination based on her conversation with the patient – a factor not recorded in any state variable.

3. The dataset $\mathcal{D}$ has been generated through a combination of algorithmic decisions and human input. For example, an engineer may tune the parameters of a recommendation algorithm based on the observed level of engagement.

If it can be demonstrated that existing OPE estimators extend to the AOPE setting with no performance drop, a new class of use-cases for Offline RL emerges. If AOPE is, in fact, verifiably harder than OPE, practitioners will know that to achieve optimal performance in Offline RL tasks, they will need to collect remarkably clean data. This paper represents our first steps towards resolving this key question.

## 2 Related Work

The OPE literature is vast. We do not attempt to provide a survey of the excellent body of work but instead refer readers to the recent work of Mou et al. [2022] and the references therein for a more comprehensive discussion. To the best of our knowledge, we are the first to study the problem of OPE under the adaptive data setting. Most existing work on OPE that we have seen makes the assumption that the data are collected iid from a single logging policy. The only exception is the work of Kallus et al. [2020] who studied OPE from multiple loggers, but they only considered the contextual bandits model and non-adaptive loggers, while we studied the RL with possibly adaptively chosen loggers.

OPE is also closely related to the average treatment effect estimation problem in the causal inference literature, but typically only only one-step decision is considered and the observational data are assumed to be iid.

## 3 Notation

Let $\Delta(\mathcal{X})$ be the set of all PMFs over $\mathcal{X}$, for $|\mathcal{X}| < \infty$. Let $[H] := \{1, ..., H\}$

A Tabular, Finite-Horizon Markov Decision Process is a tuple $(\mathcal{S}, \mathcal{A}, r, P, d_1, H)$, where $\mathcal{S}$ is the state space ($|\mathcal{S}| =: S$), and $\mathcal{A}$ is the action space ($|\mathcal{A}| =: A$). Its dynamics are governed by a nonstationary transition kernel, $P = \{P_h : \mathcal{S} \times \mathcal{A} \to \Delta(\mathcal{S})\}_{h=1}^H$, where $P_h(s'|s, a)$ is the probability of transitioning to state $s' \in \mathcal{S}$ after taking action $a \in \mathcal{A}$ from state $s \in \mathcal{S}$ at time $h \in [H]$. $r$ is a collection of reward functions $\{r_h : \mathcal{S} \times \mathcal{A} \to [-1, 1]\}_{h=1}^H$. Finally, $d_1 \in \Delta(\mathcal{S})$ is the initial state distribution of the MDP and $H$ is the horizon.

A policy, $\pi$, is a collection of maps, $\{\pi_h : \mathcal{S} \to \Delta(\mathcal{A})\}_{h=1}^H$.

Running a policy on an MDP will yield a trajectory $\tau_i \in (\mathcal{S} \times \mathcal{A} \times [-1, 1])^H$. Together, the policy and MDP induce a distribution over trajectories, as well as a Markov Chain with transitions notated as $P_h^\pi(s'|s) := \sum_a P_h(s'|s, a)\pi_h(a|s)$.

In a set of trajectories $\{\tau_i\}_{i=1}^n$, we define $n_{h,s,a}$ to be the number of visitations to $(s, a)$ at timestep $h$.

$v^\pi := \mathbb{E}_\pi[\sum_{i=1}^H r_i | s_1 \sim d_1]$ is the value of the policy $\pi$, where the expectation is over the $\pi$-induced distribution over trajectories. Similarly, $V^\pi(s) := \mathbb{E}_\pi[\sum_{i=1}^H r_i | s_1 = s]$.

$d_h^\pi(s, a)$ is defined to be the probability of $(s_h, a_h)$ occurring at time step $h$ in a trajectory distributed according to $\pi$.

## 4 Problem Formulation and Motivation

Naturally, both OPE and OL are hopeless if the logging-policy does not explore well. If the logger, $\mu$, does not visit a state that the target policy, $\pi$, visits very often, we will not be able to form an accurate estimate of the target value $v^\pi$ when doing OPE. In OL, this issue is compounded by the fact that missing *any* state may correspond to missing high-value outcomes. Thus, upper bounds on the performance of OPE or OL algorithms are given in terms of an exploration parameter, like

$$d_m = \min_{h,s,a : d_h^\pi(s,a) > 0} d_h^\mu(s, a) \tag{1}$$

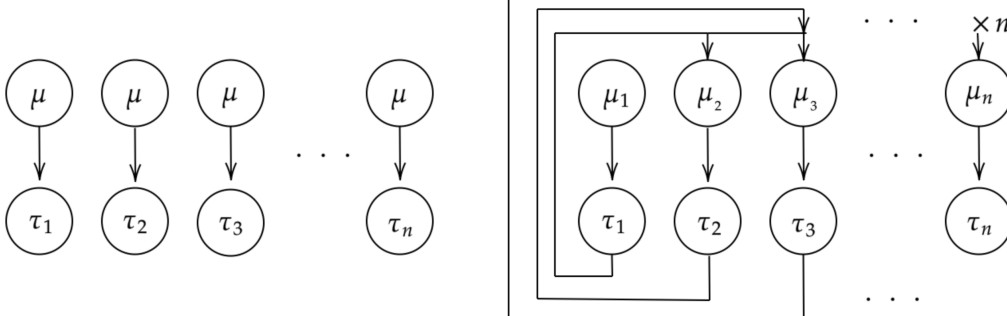

Figure 1: Non-adaptive regime (left) versus adaptive regime (right), depicted as a graphical model. We see that, in the adaptive regime, each policy depends on all previous trajectories. This induces dependence between the trajectories.

Estimators that closely match lower bounds on relevant metrics have been established in existing work, such as Duan et al. [2020]. However, the practicability of such bounds has been challenged. Xiao et al. [2022] point out that it is difficult in practice to find a logging policy with a reasonable exploration parameter. In what they consider a more realistic, "tabula rasa" case (where the logging policy is chosen without knowledge of the MDP), they show a sample complexity exponential in $H$ and $S$ to be necessary in offline learning.

Besides this limitation, we note that the motivating application of learning from existing, human-generated data remains unfulfilled in the conventional OPE setting. We would not expect these data to be identically distributed, or even independent (as the data collected in trajectory $j$ almost certainly influences future policies $\mu^{j+1}, ...$).

Thus motivated, we augment our formulation of the OPE problem to more realistically accommodate intelligent choices of logging-policy. We consider this to be middle ground between the sanguine assumptions on $d_m$ common to Yin and Wang [2020], Yin et al. [2021], Duan et al. [2020], and the assumption of total ignorance found in Xiao et al. [2022]. To this end, this paper studies two settings:

1. OPE on $\mathcal{D} = \{\tau_i \sim \mu^i\}_{i=1}^n$, where $\mu^1, ...\mu^n$ are distinct policies chosen in advance, and the trajectories unfold independently. We call this the Non-Adaptive OPE (NOPE).

2. OPE on $\mathcal{D} = \{\tau_i \sim \mu^i\}_{i=1}^n$, where $\mu^1, ...\mu^n$ are chosen adaptively. That is, $\mu^i$ may depend on the trajectories $\tau_1, ...\tau_{i-1}$. We call this Adaptive OPE (AOPE).

NOPE allows us to hedge our bets with respect to the logging policy we choose. In particular, when we derive a bound on the MSE of the model-based estimator from [Yin and Wang, 2020], it will be in terms of the minimum of the average state-action occupancy:

$$\bar{d}_m := \frac{1}{n} \min_{h,s,a} \sum_{i=1}^n d_h^{\mu^i}(s,a) \tag{2}$$

If there is some preexisting knowledge on the MDP, $\mathcal{M}$, it may be easier to propose $n$ logging policies with (2) bounded away from zero than a single logging policy with (1) bounded away from zero.

NOPE does not adequately address concerns over finding good logging policies, but rather dilutes the problem to that of finding a suite of policies that is good on average. Furthermore, it only eliminates assumptions on identical distribution, and sets aside issues raised earlier over dependence between trajectories.

Both of these problems are better addressed by AOPE. When logging policies can be tuned according to previous trajectories, there is scope for starting from "tabula rasa", and iteratively refining the logging policy as we learn about the MDP. In other words, the logger can leverage online exploration

techniques. Furthermore, by allowing arbitrary statistical dependence on previous trajectories, AOPE addresses the key scenario of learning from intradependent, manually-collected datasets.

We conclude this section by noting that minimax bounds from the non-adaptive OPE setting can be easily recovered in the following manner. For any dataset, $\mathcal{D}$, let $N := \min_{h,s,a} n_{h,s,a}$ be the number of occurrences of the least-observed $(s_h, a_h)$ pair. If we consider a revised dataset, $\mathcal{D}'$, that keeps only the first $N$ transitions out of $(s_h, a_h)$ for all $s_h, a_h$, we see that all transitions are now independent conditioned on $N$. Thus, under the conventional assumptions on exploration-parameters, the problem reduces to a generative-model type setting. In particular, Theorem 3.7 in Yin et al. [2021] implies minimax-optimal offline learning for the adaptive case. However, we do not like throwing data away in this manner, and conjecture that it should not be necessary to do so. It would be preferable to obtain bounds that adapt to the quantities $\{n_{h,s,a}\}_{h,s,a}$. To this end, we explore the extent to which *instance-dependent* bounds on estimation error can be recovered in the adaptive setting.

## 5 Method

We consider the estimator of $v^\pi$ studied in [Yin and Wang, 2020]. This boils down to computing the value of a policy under the approximate MDP defined by $(\mathcal{S}, \mathcal{A}, \hat{P}, \hat{r}, \hat{d}_1)$, with the estimators $\hat{P}$, $\hat{r}$ and $\hat{d}_1$ defined below.

That is, if $\mathcal{D} = \{\tau_1, ... \tau_n\}$, and $\tau_i = (s_1^i, a_1^i, r_1^i, ... s_H^i, a_H^i, r_H^i)$, we use plug-in estimates

$$\hat{P}_h(s'|s,a) = \frac{n_{h,s,a,s'}}{n_{h,s,a}} = \frac{1}{n_{h,s,a}} \sum_i \mathbb{1}_{\{s_h^i=s, a_h^i=a, s_{h+1}^i=s'\}} \qquad \hat{r}_h(s,a) = \frac{1}{n_{h,s,a}} \sum_{k=1}^n r_h^k \mathbb{1}_{\{s_h^k=s, a_h^k=a\}}$$

subject to these quantities being well-defined ($n_{h,s,a} \neq 0$). If $n_{h,s,a} = 0$, we can define them to be 0.

We also define $\hat{d}_1 :=: \hat{d}_1^\pi := \frac{1}{n} \sum_{i=1}^n e_{s_1^i}$ to be the plug-in estimate of $d_1$ computed from $\mathcal{D}$ (where $e_j$ is the $j$th standard basis vector in $\mathbb{R}^S$).

We then let:

$$\hat{P}_h^\pi(s'|s) = \sum_a \pi_h(a|s) \hat{P}_h(s'|s,a) \qquad \hat{r}_h^\pi(s) = \sum_a \pi_h(a|s) \hat{r}_h(s,a)$$

and iteratively compute $\hat{d}_h^\pi := \hat{P}_h^\pi \hat{d}_{h-1}^\pi$ for $h = 1, ... H$.

Finally, we form the estimate

$$\hat{v}^\pi = \sum_{h=1}^H \langle \hat{d}_h^\pi, \hat{r}_h^\pi \rangle$$

## 6 Current Results

We warm up by generalizing Yin and Wang [2020]'s bound on the MSE of $\hat{v}^\pi$ to NOPE. By following the proof of Theorem 1 in Yin and Wang [2020], and making some mild modifications, we recover a bound of the MSE of $\hat{v}^\pi$ in the Non-Adaptive OPE setting.

**Theorem 1.** *[MSE performance of $\hat{v}^\pi$ in NOPE setting] Suppose $\mathcal{D}$ is a dataset conforming to NOPE, and $\hat{v}^\pi$ is formed using this dataset. Let $\bar{d}_m$ be as defined in (2). Let $\tau_s = \max_{h,s,a} \frac{d_h^\pi(s,a)}{\frac{1}{n}\sum_i d_h^{\mu^i}(s,a)}$. Let*
$\tau_a = \max_{h,s,a} \frac{\pi_h(a|s)}{\frac{1}{n}\sum_i \mu_h^i(a|s)}$. *Then if $n > \frac{16 \log n}{\bar{d}_m}$ and $n > \frac{4H\tau_a\tau_s}{\min_{h,s} \max\{d_h^\pi(s), \frac{1}{n}\sum_i d_h^\pi(s)\}}$ we have:*

$$MSE(\hat{v}^\pi) \leq \left(1 + \sqrt{\frac{16\log n}{n\bar{d}_m}}\right) \frac{1}{n} \sum_{h,s,a} \frac{d_h^\pi(s)^2 \pi(a|s)^2}{\frac{1}{n}\sum_i d_h^{\mu^i}(s,a)} \text{Var}[r_h^{(1)} + V^\pi(s_{h+1}^{(1)})|s_h^{(1)} = s, a_h^{(1)} = a]$$

$$+ O(\tau_a^2 \tau_s H^3/n^2 \bar{d}_m)$$

As a corollary, consider a "quasi-adaptive" data collection process, where each logging-policy, $\mu^i$, is run twice, generating i.i.d. $\tau_i$ and $\tau_i'$. Suppose future logging-policies $\mu^{j>i}$ are chosen by some

algorithm $\mathcal{E}$ depending on $\tau_i$ but not $\tau_i'$. We can use the same $\hat{v}^\pi$-estimator to perform OPE with $\mathcal{D}_{shadow} = \{\tau_i'\}$, as long as the estimator doesn't touch $\mathcal{D} = \{\tau_i\}$. If we assume that average exploration is sufficient w.h.p. over the execution of $\mathcal{E}$, we can the bound MSE in this quasi-adaptive case using Theorem 1, the fact that the $\{\tau_i'\}$ are mutually independent conditioned on $\{\mu^i\}$, and the tower rule.

**Corollary 2.** *Let $\mathcal{E}$ be the algorithm described in the paragraph above. Assume that with high probability ($\geq 1 - \delta$), the policies $\mu^1, ...\mu^n$ generated by $\mathcal{E}$ satisfy $\frac{1}{N}\sum_i d_h^{\mu^i}(s,a) \geq \bar{d}_m$ for all $s \in S$ and $a \in A$, and for some $\bar{d}_m$. Then*

$$MSE(\hat{v}^\pi) \leq (1-\delta)(*) + H^2\delta$$

*where $(*)$ is the bound on the MSE of the estimator in the non-adaptive case from Theorem 1.*

We now turn our attention towards quantifying $\hat{v}^\pi$'s performance on AOPE. We first describe a high-probability, uniform error bound in terms of the number of visitations to each $(s_h, a_h)$ tuple.

**Theorem 3** (High-probability uniform bound on estimation error in AOPE). *Suppose $\mathcal{D}$ is a dataset conforming to AOPE, and $\hat{v}^\pi$ is formed using this dataset. Then, with probability at least $1 - \delta$, the following holds for all policies $\pi$.*

$$|\hat{v}^\pi - v^\pi| \leq K \sum_{h=1}^{H} \sum_{s,a} H d_h^\pi(s,a) \sqrt{\frac{S \log \frac{HSAn}{\delta}}{n_{h,s,a}}}$$

*where $n_{h,s,a}$ is the number of occurrences of $(s_h, a_h)$ in $\mathcal{D}$, and $K$ is an absolute constant.*

This translates to the following worst-case bound, which underperforms the minimax-optimal bound (over deterministic policies) implied by Yin et al. [2021] by a factor of $\sqrt{H}$.

**Corollary 4** (High-probability uniform bound on estimation error in AOPE). *Suppose that $\mathcal{D}, \hat{v}^\pi$ are as in Theorem 3. Then with probability $1 - \delta$, we have that*

$$\sup_\pi |\hat{v}^\pi - v^\pi| \leq O(H^2 \sqrt{\frac{S \log HSAn/\delta}{n\bar{d}_m}})$$

The proof of Theorem 3 follows by a simulation lemma-type expansion of the error, which leads to a dominant term of the form $\sum_h \mathbb{E}_{s_h,a_h \sim \pi, \mathcal{M}}[(\hat{P}_{h+1}(\cdot|s_h, a_h) - P_{h+1}(\cdot|s_h, a_h))^T \hat{V}_{h+1}^\pi]$, and smaller terms governed by $\hat{r}$ and $\hat{d}_1$. In order to get around the issue of dependence between trajectories, we cover all possible number of occurrences of each $(s_h, a_h)$ across trajectories while applying concentration, leading to to the $HSAn$ term inside the logarithm.

We also give a high-probability, instance-dependent, *pointwise* bound, which is suboptimal by a factor of $\sqrt{H}$ when translated into a worst-case bound. In the pointwise case, we are able to shave off a $\sqrt{S}$ in the asymptotically dominant term.

**Theorem 5** (Instance-dependent pointwise bound on estimation error in AOPE). *Fix a policy $\pi$, suppose $\mathcal{D}$ is a dataset conforming to AOPE, and $\hat{v}^\pi$ is formed using this dataset. Assume that with probability $\geq 1 - \frac{\delta}{2}$, $n_{h,s,a} \geq n\bar{d}_m$ for all $h, s, a$ for some $\bar{d}_m > 0$. Then with probability at least $1 - \delta$, we have:*

$$|\hat{v}^\pi - v^\pi| \leq O(\sum_{h=1}^{H} \sum_{s,a} d_h^\pi(s,a) \sqrt{\frac{\text{Var}_{s' \sim P_{h+1}(\cdot|s,a)}[V_{h+1}^\pi(s')] \log \frac{HSAn}{\delta}}{n_{h,s,a}}} + \frac{H^2 S \log \frac{HSAn}{\delta}}{n\bar{d}_m})$$

The above translates into the following worst-case bound.

**Corollary 6** (Worst-case pointwise bound on estimation error in AOPE). *Fix a policy $\pi$, suppose $\mathcal{D}$ is a dataset conforming to AOPE, and $\hat{v}^\pi$ is formed using this dataset. Then with probability at least $1 - \delta$, we have:*

$$|\hat{v}^\pi - v^\pi| \leq O(\sqrt{\frac{H^3 \log HSAn/\delta}{n\bar{d}_m}} + \frac{H^2 S \log \frac{HSAn}{\delta}}{n\bar{d}_m})$$

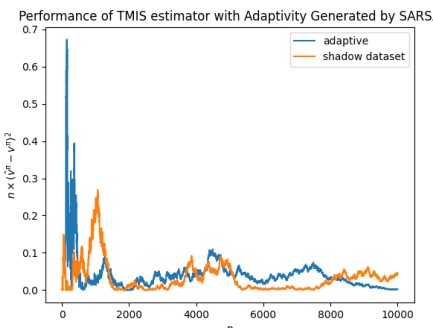
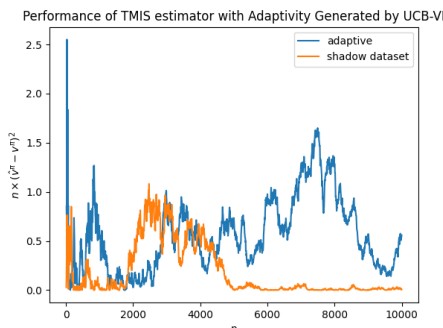

Figure 2: Single-run results of the simulations described in Section 7. On the $x$-axis, number of trajectories used by $\hat{v}^\pi$. On the $y$-axis, the square-error scaled by $n$.

Inspired by Azar et al. [2017], Theorems 5 is proved by applying concentration inequalities (with the same covering trick as Theorem 3) to $(\hat{P}_{h+1} - P_{h+1})V_{h+1}^\pi$ and $(\hat{P}_{h+1} - P_{h+1})(\hat{V}_{h+1}^\pi - V_{h+1}^\pi)$ separately, instead of $(\hat{P}_{h+1} - P_{h+1})\hat{V}_{h+1}^\pi$. In order to treat the dominant term, we use Bernstein's inequality. To recover the worst-case bound in the corollary, we analyze the variance term with the canonical equality $\sum_h E_\pi[\text{Var}_{s' \sim P(\cdot|s,a)}[V_h^\pi(s')]] = \text{Var}_\pi[\sum_h r_h] - \sum_h \mathbb{E}_\pi[\text{Var}[\mathbb{E}[r_h + V_{h+1}^\pi(s')|s,a]]] = O(H^2)$.

## 7 Numerical Simulations

In Section 4, we stated our conjecture that learning from adaptively collected datasets should not be harder than learning from mutually independent datasets. In this section, we present the early results of simulations designed to study this question.

The simulations were conducted on a toy MDP with two states, two actions, and nonstationary dynamics, and performed as follows. Fix a type of adaptive algorithm, $\mathcal{Z}$ (we used the well-known exploration algorithms UCB-VI [Azar et al., 2017] and SARSA), and a target policy, $\pi$. [1]

- Collect a dataset of $k$ trajectories, $\mathcal{D}$, by running $\mathcal{Z}$. Record the policies $\mathcal{Z}$ uses in $\Pi_\mathcal{Z}$
- Collect a "shadow" dataset, $\mathcal{D}_{shadow}$ by running each of the $k$ policies in $\Pi_\mathcal{Z}$
- For both datasets, and for a range of $n \in \{1, ...k\}$, estimate and $\hat{v}^\pi$ using the first $n$ trajectories in $\mathcal{D}$, and do the same for $\mathcal{D}_{shadow}$

At a glance, Figure 2 suggests that the adaptive dataset does not suffer a performance hit in either adaptivity regime. However, this data is noisy, and we intend to gather more extensive empirical evidence for this conclusion in future.

## Next Steps

On the theoretical side, the instance-dependent results from Theorems 3 and 6 fail to recover the correct minimax behavior. This is because we could not salvage the martingale structure leveraged across timesteps in [Yin et al., 2021]. However, based on our early simulations and our intuition that throwing away data should not help us learn, we believe this gap to be an artifact of our analysis. As future work, we intend to recover better bounds by more carefully analyzing the structure of the MDP, or else demonstrate (by means of a lower bound) that the estimator $\hat{v}^\pi$ is worse at AOPE than OPE.

On the experimental side, we plan on carrying out more detailed simulations than those presented in Section 7. Empirically observing the behavior of certain error metrics under various adaptivity regimes may illuminate theoretical aspects of the problem (especially concerning how adaptivity can give rise to estimation bias for certain policies, an extension from Multiarm Bandits to RL of the phenomena studied in [Shin et al., 2019]).

---

[1]We used a data-dependent $\pi$

We believe that theoretically clarifying the extent to which OPE methods carry over to the AOPE setting is a step towards making offline RL a yet more convincing candidate for real-world decision problems.

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
