# OpenReview forum: "Offline Policy Evaluation for Reinforcement Learning with Adaptively Collected Data"
_NeurIPS.cc/2022/Workshop/Offline_RL — Offline RL Workshop NeurIPS 2022_

### Official Review · Reviewer_B8ea · 2022-10-14
**A reasonable theoretical waypoint**

**Rating:** 7
**Confidence:** 3

**Review:**

Authors study the deviation properties of an offline plug-in estimator of transition and reward distributions in tabular MDP,   The main innovation is considering predictable logging policy sequences.

Unfortunately, the format of the workshop precludes including proofs which makes it difficult to assess correctness.  The reviewer will assume correctness.  Under this assumption, the "next steps" are reasonable, especially understanding whether the $\sqrt{H}$ factor is fundamental or an artifact of the analysis.

> If there is some pre-existing knowledge on the MDP, M, it may be easier to propose n logging
policies with (2) bounded away from zero than a single logging policy with (1) bounded away from
zero.

for NOPE, is there a distinction between playing the uniform stochastic mixture policy over the specified policy sequence and applying equation (1), vs. playing the policy sequence in order and applying equation (2)?

---

### Official Review · Reviewer_KXAv · 2022-10-18
**Good theoretical analysis of the OPE problem, however no evaluation of the presented techniques.**

**Rating:** 6
**Confidence:** 3

**Review:**

The paper points out the problems with some assumptions made when dealing with offline datasets in RL, such as independence and quality of exploration in the trajectories collected by a logging policy. The paper studies two settings of the logging policy choice, one where there are multiple independent logging policy and one where the logging policy adapts to the collected trajectories. The latter is shown to be better suited to the common scenarios in offline learning.

The paper performs a rigorous theoretical analysis of the offline policy evaluation problem. However, the developed techniques are not evaluated in any experiments.